# How Well Can LLMs Synthesize Molecules? An LLM-Powered Framework for Multi-Step Retrosynthesis

## Abstract

Predicting retrosynthesis routes is a fundamental challenge in chemistry, involving the design of a sequence of chemical reactions to synthesize a target molecule from commercially available starting materials. With a rapidly growing interest in using large language models for planning, this work introduces an LLM-powered framework for multi-step retrosynthesis. Our framework employs molecular-similarity-based retrieval-augmented generation (RAG) to generate an initial retrosynthesis route, which is then iteratively refined through expert feedback. The use of molecular-similarity-based RAG improves reaction round-trip validity from 24.42% to 51.64% compared to GPT-4 with representative routes. With further refinement, the validity increases to 89.81%, resulting in an overall route validity of 79.5% with a perfect query success rate, comparable to traditional methods. Our framework offers a flexible, customizable approach to retrosynthesis, and we present a comprehensive analysis of the generated routes along with promising future research directions in LLM-driven multi-step retrosynthesis.

## 1 Introduction

In recent years, large language models (LLMs) have demonstrated remarkable capabilities in planning tasks across various domains, including robotics, entertainment, and scientific problem-solving (Hu et al., 2024; Prasad et al., 2024; Sun et al., 2023; Zhang & Lu, 2024; Tan et al., 2024; Zhou et al., 2023; Yu et al., 2024a; Shinn et al., 2023; Yao et al., 2023; Lu et al., 2023; Trinh et al., 2024). These approaches leverage LLMs with advanced architectures pretrained on vast datasets, employ well-structured prompts and external expert tools, and integrate them into carefully designed system pipelines to generate coherent plans and strategies. In many cases, LLMs have achieved results comparable to, or even surpassing, traditional methods.

Finding a synthesis route for a given molecule is a pivotal challenge in chemistry with earliest efforts traces back to the 1960s(Corey, 1967). Existing approaches address this from a retrosynthesis perspective by framing the problem as a navigating task through an AND-OR tree rooted on the target molecule node, as illustrated in Figure A1, a process known as retrosynthesis planning, which will be discussed in detail in later sections. Similar to planning in other domains, retrosynthesis requires both high-level reasoning to guide the overall process and a deep understanding of chemistry to ensure that each step involves feasible actions. Given the demonstrated success of LLMs in various fields, a compelling question arises: ***Can LLMs be effectively applied to discover retrosynthesis pathways for molecules?***

In this work, we introduce a novel methodology for retrosynthesis route generation, where the entire route is generated and refined holistically, rather than being iteratively expanded step-by-step as in traditional approaches. This approach leverages the inherent ability of LLMs to generate complex sequences, allowing for more flexibility and creativity in the initial prediction with guidance from reference routes provided by the molecular-similarity-based RAG module. By incorporating RAG, our method guides LLMs to reference retrosynthesis routes from structurally similar molecules, providing a more informed basis for initial predictions. This is particularly important, as structurally similar molecules often exhibit analogous reactivity patterns, making them valuable references. We propose an iterative refinement framework, ensuring that the generated route can be adjusted and optimized as a whole. The iterative refinement process is driven by expert models, which offer targeted feedback to improve the accuracy and feasibility.

Furthermore, the use of local knowledge bases ensures that the generated routes remain chemically viable and aligned with existing knowledge, ultimately enhancing the robustness and practicality of the retrosynthesis plan.

**Our contributions:** We present a novel LLM-powered framework for multi-step retrosynthesis route generation, designed to generate and refine routes holistically rather than through iterative expansion. Using this framework, we evaluate both closed-source and open-source LLMs, including GPT-4-turbo, Claude-3-Haiku, and Deepseek-V2.5. Our study identifies several shortcomings and vulnerabilities across these models, highlighting key design considerations and pointing to promising future research directions. Collectively, this paper offers three main contributions: 1) A new framework for multi-step retrosynthesis powered by LLMs; 2) a comprehensive quantitative evaluation of LLM performance in retrosynthesis route generation; and 3) new insights into LLM behaviours that influence retrosynthesis route generation.

## 2 PRELIMINARY

### 2.1 LLMs WITH IN-CONTEXT LEARNING FOR CHEMISTRY

LLMs pretrained on general knowledge corpora struggle with even basic chemistry tasks, such as converting between the Simplified Molecular Input Line Entry System (SMILES) and International Union of Pure and Applied Chemistry (IUPAC) names (Castro Nascimento & Pimentel, 2023), let alone more complex challenges like retrosynthesis route generation.

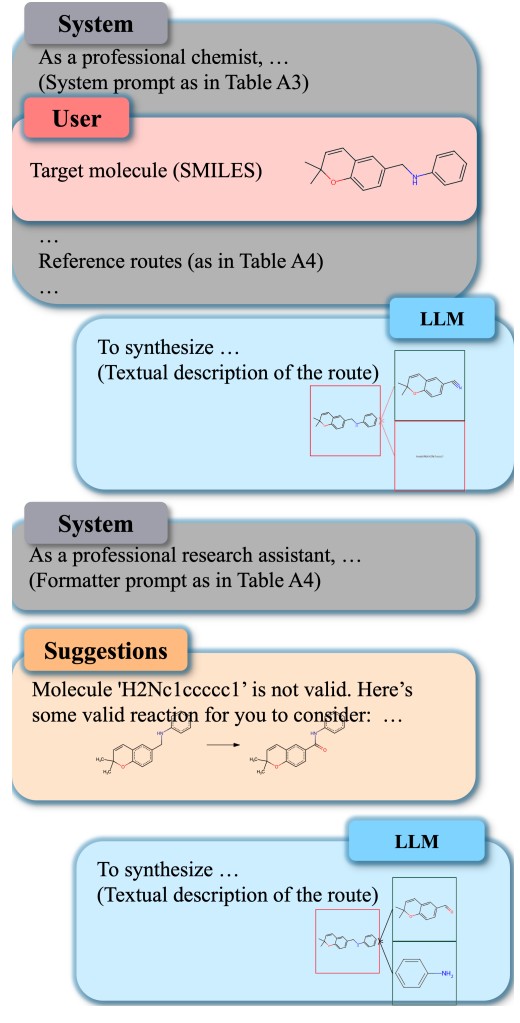

Figure 1: An example of valid retrosynthesis route generated by our framework.

Fine-tuning a pretrained LLM with parameters $\theta$ on a domain-specific dataset $\mathcal{D}_{\text{domain}} = \{(x_i, y_i)\}_{i=1}^{N}$, to obtain updated parameters $\theta^* = \arg\min_\theta \frac{1}{N} \sum_{i=1}^{N} \mathcal{L}(\theta; x_i, y_i)$—where $\mathcal{L}$ is the loss function—can help address this issue, as demonstrated in (Fang et al., 2023; Zeng et al., 2022; Liu et al., 2023c). However, this approach presents three significant challenges: (1) it is computationally expensive, requiring substantial resources, (2) it is data-intensive, with a scarcity of high-quality retrosynthesis route datasets in the community, and (3) it lacks flexibility and cannot adapt to new knowledge without the integration of additional continual learning modules.

To overcome these limitations, an alternative approach is to leverage in-context learning (ICL). Opposed to the fine-tuning paradigm, ICL allows LLMs to adapt to specialized tasks by incorporating relevant examples directly into the input prompt at inference time, i.e. $\hat{y}_{ICL} = P(x, \{x_j, y_j\}|\theta)$, where $\{x_j, y_j\} \subset \mathcal{D}_{\text{domain}}$. This approach offers a more flexible and efficient solution, without the need for additional training or large volumes of annotated data. Ramos et al. (2023); Li et al. (2023); Edwards et al. (2023) adapted this approach for chemistry-related tasks.

### 2.2 RETROSYNTHESIS PLANNING

A retrosynthesis route $R_t$ is a sequence of chemical reactions $[r_1, r_2, \ldots, r_n]$ designed to synthesize a target molecule $t$. Each reaction must involve chemically feasible molecules, denoted as $M$, and all intermediate materials used in the synthesis, denoted as $I$, must be synthesizable from a

set of commercially available building-block materials, denoted as $B$. Given the vast size of $B$ (23.1M molecules in *eMolecules*[1] as of 2019), forward reasoning from $B$ to $t$ is inefficient. A more commonly adopted methodology is planning-based, which begins with decomposing $t$ with a predicted reaction $r_1$ to form $I_0$, where $I_0$ is the set or reactants suggested by $r_1$. At the $i$th step, planning-based approaches select molecules $m \in I_i$ to decompose with predicted single-step retrosynthesis reaction $r_i$, thereby expanding both the set of reactions $R$ and the set of intermediates $I$. This iterative process stops when all intermediates $i \in I$ are decomposed to molecules present in $B$. This process is typically abstracted as navigating through an AND-OR tree, comprising selection, expansion, and update phases, as illustrated in Figure A1.

In the planning-based paradigm, explicit value functions $f_V : M \mapsto \mathbf{R}$, which map an intermediate molecule to a score, are necessary to select the appropriate $m \in I$ for further decomposition.

## 2.3 Rethinking Multi-Step Retrosynthesis

Our vision is to generate complete retrosynthesis routes, $R_t$, in a single pass, without relying on the iterative selection-expansion phases typical of traditional methods. Although generated $R_t$ may not represent the final route, it can be iteratively refined based on user preferences or expert knowledge as holistically, with the assistance of LLMs. This process allows for flexible adjustments, as modifications can be applied to a complete route rather than to partial sequences. In line with Strieth-Kalthoff et al. (2024)'s vision of utilising both experts knowledge and data-intensive models, this approach fosters a more dynamic, user-driven process while improving the reliability and quality of the retrosynthesis route by enabling direct user feedback and simplifying optimization.

## 3 Utilizing LLM for Retrosynthesis Route Generation

**Pipeline** Building on our vision of holistic retrosynthesis route generation, we propose a retrosynthesis route generation framework that employs molecular-similarity-based RAG to generate an initial, possibly flawed retrosynthesis route $R_{t_0}$. This route is then iteratively refined using feedback from expert models and local knowledge databases to form the final prediction $R_t$. The overall pipeline is illustrated in Figure 2, with pseudocode provided in Algorithm 1. Our approach comprises four key components: a molecular-fingerprint-based RAG module, an LLM-backed formatter module, an expert-powered feedback module, and a local knowledge database module. These components will be discussed in detail in the subsequent sections.

**Molecular-similarity-based Retrieval-Augmented Generation** Unlike RAG applications in the natural language domain where the similarity metric is calculated on the embedding of texts, we utilize the similarity of molecular fingerprints to find similar molecules and provide their synthesis routes to the LLM for an initial generation. More specifically, a vector database that uses the molecular fingerprint as embedding and Tanimoto similarity as a similarity metric is applied for the retrieval. We begin by filtering out routes associated with the top five most similar molecules, each having a Tanimoto similarity greater than 0.5 in the database; these are designated as the retrieved routes. If no similar routes are found, we instead rely on representative routes. To identify these, we calculate the route fingerprint as the sum of the reaction fingerprints, following the method proposed by Schwaller et al. (2021). We then apply the Density-based spatial clustering of applications with noise (DBSCAN) clustering algorithm (Ester et al., 1996) to these route fingerprints, generating five distinct reaction clusters. Within each cluster, the route corresponding to the molecule nearest to the cluster centroid is selected as the representative route. Since directly restricting the output format may lead to performance degradation as suggested by Tam et al. (2024), we provide those reference routes as textual descriptions in the prompt as shown in Figure 3c. The textual description is generated using a rule-based method, highlighting route connectivity information. A Chain-of-Thought(CoT)-like (Wei et al., 2023) prompt with reference routes and specific instructions to generate textual descriptions of routes is fed into the LLM to guide the generation, the prompt template is included in Table A3.

**LLM-backed Formatter** LLM-backed formatter converts the textual description of a retrosynthesis route into a format readable to human experts and compatible with expert models. Particularly,

---

[1]https://www.emolecules.com/products/building-blocks

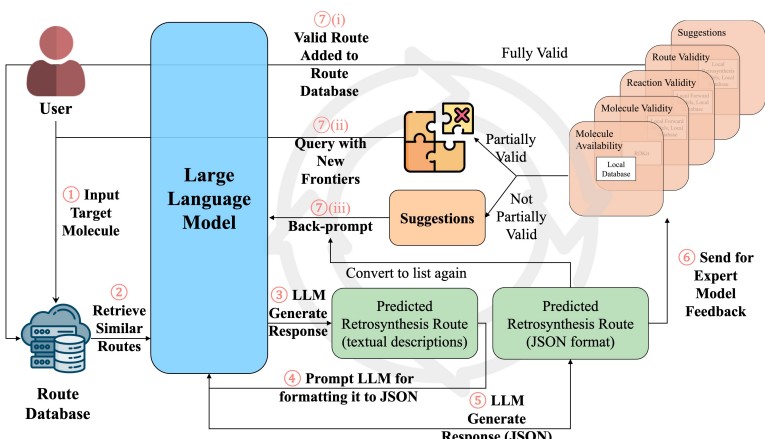

Figure 2: Pipeline for generating retrosynthesis routes using LLMs. The process begins with the user inputting a target molecule (Step ①). The system then queries a route database to retrieve similar retrosynthesis routes (Step ②), which guides the LLM in generating an initial prediction (Step ③). The predicted retrosynthesis route is converted to JSON format with help from LLMs (Steps ④ and ⑤). The JSON-formatted route is sent for expert model feedback (Step ⑥) to verify the validity of the route, assessing various criteria such as route validity, reaction validity, and molecule availability. If the route is fully valid, it is added to the route database (Step ⑦(i)). In cases where the route is partially valid, the pipeline queries new frontiers and back-prompts the LLM to refine the prediction (Steps ⑦(ii) and ⑦(iii)). Steps ③–⑦ are repeated until a fully valid retrosynthesis route is generated or a predefined budget is reached.

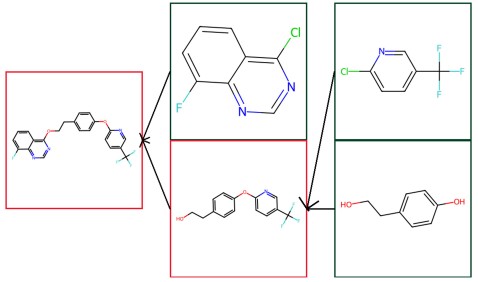

(a) Visualized example route.

```
[
    Fc1ccc2c(OCCc3ccc(Oc4ccc(C(F)(F)F)
        cn4)c3)nnc2
            >>
        Fc1ccc2c(Cl)ncnc12.
        OCCC1ccc(Oc2ccc(C(F)(F)F)cn2)cc1,

    OCCC1ccc(Oc2ccc(C(F)(F)F)cn2)cc1
            >>
        FC(F)Fc1ccc(Cl)c1.
        OCCCC1ccc(O)cc1
]
```

(b) Example route represented as a list of reaction SMILES

The target molecule, Fc1cccc2c(OCCc3ccc(Oc4ccc(C(F)(F)F)cn4)cc3)ncnc12, can be synthesized by reacting the following precursor molecules: Fc1cccc2c(Cl)ncnc12 and OCCc1ccc(Oc2ccc(C(F)(F)F)cn2)cc1.

To synthesize Fc1cccc2c(Cl)ncnc12, which is a precursor reactant in the above reaction Fc1cccc2c(OCCc3ccc(Oc4ccc(C(F)(F)F)cn4)cc3)ncnc12>>Fc1cccc2c(Cl)ncnc12.OCCc1ccc(Oc2ccc(C(F)(F)F)cn2)cc1, since it is commercially available as a building block material, so no further synthesis is needed.

To synthesize OCCc1ccc(Oc2ccc(C(F)(F)F)cn2)cc1, which is a precursor reactant in the above reaction Fc1cccc2c(OCCc3ccc(Oc4ccc(C(F)(F)F)cn4)cc3)ncnc12>>Fc1cccc2c(Cl)ncnc12.OCCc1ccc(Oc2ccc(C(F)(F)F)cn2)cc1, can be synthesized by reacting the following precursor molecules: FC(F)(F)c1ccc(Cl)nc1 and OCCc1ccc(O)cc1.

To synthesize FC(F)(F)c1ccc(Cl)nc1, which is a precursor reactant in the above reaction OCCc1ccc(Oc2ccc(C(F)(F)F)cn2)cc1>>FC(F)(F)c1ccc(Cl)nc1.OCCc1ccc(O)cc1, since it is commercially available as a building block material, so no further synthesis is needed.

To synthesize OCCc1ccc(O)cc1, which is a precursor reactant in the above reaction OCCc1ccc(Oc2ccc(C(F)(F)F)cn2)cc1>>FC(F)(F)c1ccc(Cl)nc1.OCCc1ccc(O)cc1, since it is commercially available as a building block material, so no further synthesis is needed.

(c) Example route represented as textual descriptions.

```
{
    'molecule':
        'Fc1cccc2c(OCCc3ccc(Oc4ccc(C(F)(F)F)cn4)cc3)ncnc12',
    'children': [
        {
            'molecule': 'Fc1cccc2c(Cl)ncnc12',
            'children': []
        },
        {
            'molecule': 'OCCc1ccc(Oc2ccc(C(F)(F)F)cn2)cc1',
            'children': [
                {
                    'molecule': 'FC(F)(F)c1ccc(Cl)nc1',
                    'children': []
                },
                {
                    'molecule': 'OCCc1ccc(O)cc1',
                    'children': []
                }
            ]
        }
    ]
}}
```

(d) Example route represented as a tree in JSON.

Figure 3: Different sequential representations of retrosynthesis routes

the route is formatted into a synthesis tree in JSON format as shown in Figure 3d. This formatter is also in charge of completing or proofreading the generated route. For instance, the generated description may contain a category of molecules instead of a specific molecule, the formatter is

prompted to provide a concrete example with its knowledge. Moreover, the description may contain IUPAC names while expert models require the SMILES string, hence an external tool is provided to the formatter to query PubChem for this naming conversion.

**Feedback from Experts** The synthesis tree is evaluated by a series of expert models for feedback, ranging from molecule-wise to reaction-wise, and finally route-wise. Specifically, LLM-generated routes are first analyzed for SMILES validity and node availability. This is done using RDKit and cross-referencing with the molecule-level availability database. Any invalid SMILES or unavailable leaf nodes are flagged in the feedback. Next, forward reaction prediction models are applied to assess whether the proposed reactants can synthesize the target products. If the predicted reactions are infeasible, reaction-level database and retrosynthesis prediction models are employed to generate suggestions for refining the LLM-generated routes. Route-wise feedback focuses on identifying disconnected synthesis pathways or incorrect final products. For forward reaction prediction, we utilize MolecularTransformer (Schwaller et al., 2019), a template-free approach, and LocalTransform (Chen & Jung, 2022), a template-free approach. Retrosynthesis predictions are performed using LocalRetro (Chen & Jung, 2021), a template-based framework, and a one-step retrosynthesis MLP model described in (Chen et al., 2020). Finally, a rule-based system integrates input from local expert models to address various types of errors, as outlined in Table A2. Note that those expert models, while performing well on individual datasets, often fail to cope with out-of-distribution datasets as pointed out by Yu et al. (2024b). Hence, we also build a web-based interface to take advice from human experts as illustrated in Figure A3 While the framework has the capability to incorporate feedback from human experts, this feature has not been tested in our experiments.

**Local Knowledge Databases** Following the feedback module, the local knowledge databases are organized into three levels of granularity: molecule level, reaction level, and route level. The molecule-level database contains 23 million commercially available molecules. The reaction-level database is initialized with all retrosynthesis reactions from the training dataset and is expanded using the results from retrosynthesis prediction models. The route database starts with all routes and sub-routes from the training dataset and is further expanded by incorporating LLM-generated routes or sub-routes once they have passed all expert scrutinies. Sub-routes refer to the synthesis pathways of intermediate products.

## 4 BENCHMARKING LLMs FOR RETROSYNTHESIS ROUTE GENERATION

### 4.1 EXPERIMENTAL SETUP

- **Dataset** We perform experiments on retro* dataset introduced by Chen et al. (2020) and evaluate the performance of our framework on a slightly harder subset of its test set as shown in Table A1. Experiments on routes extracted from the Pistachio dataset [2] are presented in A.10.

- **Baseline** We select Retro* (Chen et al., 2020) and EG-MCTS (Hong et al., 2023) as our baselines. We limit the number of expansions for baseline methods to 5 as only 5 suggested reactions are given to LLMs and the iteration budget is set to 500 as in (Chen et al., 2020). We also fine-tuned ChemDFM-V1.5-8B (Zhao et al., 2024) to generate retrosynthesis routes as the baseline of fine-tuning approach, more details on the fine-tuning process can be found in Table A6.

- **Metrics** We employ a diverse set of both text-based and chemistry-based metrics to evaluate the quality of the generated retrosynthesis routes, considering their similarity to reference routes as well as their chemical feasibility, including: 1) *Query Success Rate*, the percentage of instances where a route is successfully generated for a target molecule, regardless of quality; 2) *ROUGE* (Recall-Oriented Understudy for Gisting Evaluation), introduced by Lin (2004), measures the overlap of textual elements between generated and reference outputs. Specifically, we report the ROUGE-1 score between the string representation of the ground truth retrosynthesis tree and the generated retrosynthesis tree, both formatted in JSON, as illustrated in Fig. 3d. 3) *BLEU* (Bilingual Evaluation Understudy), proposed by Papineni et al. (2002), evaluates the precision of n-gram overlaps between the generated and reference outputs. Similar to ROUGE, we calculate the BLEU score using the string representations of the retrosynthesis trees. 4) *Exact Match* evaluates the percentage of instances where the generated retrosynthesis tree is identical to the reference tree. To

[2]https://www.nextmovesoftware.com/pistachio.html

ensure consistency, retrosynthesis trees are canonicalized by standardizing molecule SMILES and sorting precursor molecules alphabetically. 5) *Molecular Validity*, the percentage of chemically feasible molecules in the generated routes; 6) *Route Validity*, where a route is valid if the final product is the target molecule, all molecules are chemically feasible, all reactions are round-trip (RT) valid, and all leaf nodes are commercially purchasable; 7) *Avg. Route Length*, the average number of reactions used in valid routes to synthesize target molecules. We extend the original definition of round-trip (RT) validity from Schwaller et al. (2020), defining it as the ability to synthesize the same products from the given reactants. We use an ensemble approach combining database, template-free, and template-based models to evaluate RT accuracy. A reaction is considered RT valid if it exists in the reaction database (of reaction from all splits) or is classified as top-5 RT valid by either a template-free or template-based model. Though several methods are applied for RT validation, it remains flawed without experimental verification.

- **LLMs** We conducted experiments using the following models: GPT-4-turbo, Claude-3.5-haiku, and Deepseek-V2.5 [3]. Deepseek-V2.5 is a representative of open-sourced LLM but we use its API version. The pricing for these models varies significantly: GPT-4-turbo costs \$10 per million input tokens and \$30 per million output tokens, Claude-3-Haiku is priced at \$0.25 per million input tokens and \$1.25 per million output tokens, and Deepseek-V2.5 charges \$0.14 per million input tokens and \$0.25 per million output tokens [4].

- **Configuration** We set the iteration budget to 5. In each iteration, local experts can provide up to 5 valid single-step reactions for the LLMs to consider when no matching reaction is found in the database. Unless otherwise specified, the same set of LLMs is used for both generation and formatting tasks. In case of an unexpected error, the LLM is allowed up to 3 retry attempts.

## 4.2 MAIN RESULTS

| Metric / Method | Query Success Rate ↑ | Rouge ↑ | Bleu ↑ | Exact Match ↑ | Molecule Validity ↑ | Route Validity ↑ | Average Length of Valid Route ↓ |
|---|---|---|---|---|---|---|---|
| GPT-4-turbo | 100.00% | 0.7649 | 0.6742 | 14.50% | 99.21% | 79.50% | 3.30 |
| Claude-3-haiku | 100.00% | 0.7279 | 0.5493 | 15.00% | 93.52% | 62.50% | 2.99 |
| Deepseek-V2.5 | 100.00% | 0.7399 | 0.4428 | 17.00% | 86.76% | 67.50% | 2.75 |
| Finetuned w RAG | 100.00% | 0.6692 | 0.6724 | 5.00% | 98.48% | 26.50% | 1.81 |
| Retro* | 98.00% | 0.7499 | 0.7825 | 30.50% | 100.00% | 83.00% | 2.58 |
| EG-MTCS | 99.0% | 0.7447 | 0.7554 | 27.50% | 100.0% | 82.50% | 2.34 |
| Ground truth | 100.00% | 1.0000 | 1.0000 | 100.00% | 100.00% | 100.00% | 3.12 |

Table 1: Comparison of our proposed methods backed by different LLMs with the baseline. Molecule validity for Retro* is measured based on their predicted routes, while in our proposed framework, it is measured on the final predicted routes. The same applies to product similarity and route round-trip validity. The average route length is reported only for round-trip valid routes.

We report the analysis on the route-level metrics of our proposed framework using different LLMs against baseline and ground truth as shown in Table 1. We report several key findings as follows.

**Our method is capable of generating round-trip valid retrosynthesis routes.** In our proposed framework, specifically, the one backed by GPT-4-turbo, achieves a route round-trip validity of 79.5%, which is comparable to the baseline Retro* method at 83.0%. Since our feedback and suggestions primarily target round-trip validity, this demonstrates the effectiveness of our proposed refinement scheme, which could be extended to other metrics with proper suggestions provided. Additionally, the GPT-4-turbo-backed approach successfully identified several round-trip valid routes that Retro* was unable to generate as shown in case analysis in Section A.6.

**Finetuned LLMs Understand Formatting but Struggle with Retrosynthesis Planning Nuance.** While finetuned LLMs achieve competitive Rouge and Bleu scores compared to our proposed GPT-4-turbo-based approach (0.6692 vs. 0.7649 for Rouge and 0.6724 vs. 0.6742 for Bleu), they fail to generate as many valid retrosynthesis routes. This suggests that while these models effectively learn the structure and formatting of retrosynthesis trees during supervised fine-tuning, they lack the deeper understanding required to capture the nuanced decision-making processes involved in

---

[3]We use gpt-4-turbo-2024-04-09, claude-3-haiku-20240307 and refer them as gpt-4-turbo and claude-3-haiku for brevity.

[4]As of September 2024.

retrosynthesis planning. Even chemistry-aware models, such as ChemDFM, demonstrate limited ability to fully emulate these subtleties, emphasizing the added value of our approach.

**Traditional planners still excel in identifying more reliable routes.** While our framework shows promising results in some metrics, traditional planners such as Retro* still hold a clear advantage in terms of molecule validity and route efficiency. Retro* achieves 100% molecule validity, meaning that every molecule in the generated routes is chemically valid, whereas GPT-4-turbo lags slightly behind at 99.21%. Additionally, Retro* consistently produces shorter routes, with an average valid route length of 2.58, compared to the longer average of 3.30 steps in GPT-4-turbo. This suggests that traditional planners not only maintain better accuracy in generating valid molecules but also find more concise and efficient retrosynthesis routes. It is worth noting that while our current framework primarily focuses on route validity, introducing explicit feedback regarding route length could further enhance the performance of our framework—a direction we reserve for future work. A detailed analysis of the source of the performance differences will be presented in the following text.

### 4.3 INSIGHTS FROM THE EXPERIMENTS

**RAG Quality Matters.** We plot the reaction-level top-5 round-trip validity of each route generated in the first iteration against two factors: the synthesis difficulty of the molecule, measured by its SA score, and the quality of the retrieved examples, measured by the average molecular similarity between the target molecule and the retrieved examples, as shown in Figure 4a. Additionally, we compute the correlation coefficient and p-value using the Spearman rank correlation test. The results indicate a weak but statistically significant positive correlation between the round-trip validity of the generated routes and the quality of retrieval. However, no meaningful relationship is observed between the round-trip validity and the synthesis difficulty during the first iteration. This suggests that the performance of our proposed approach could be further enhanced by expanding the route database with additional relevant examples for retrieval. We also conducted an ablation experiment

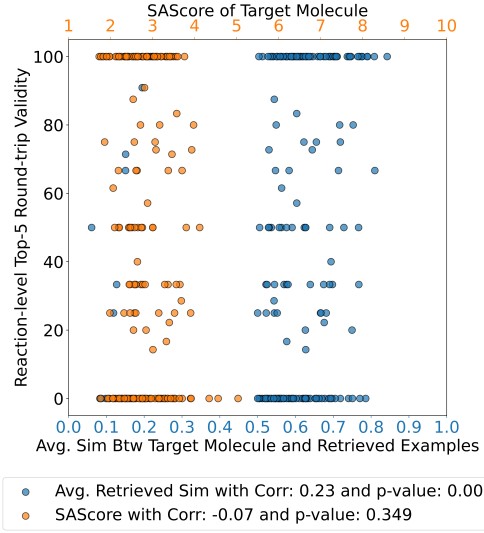

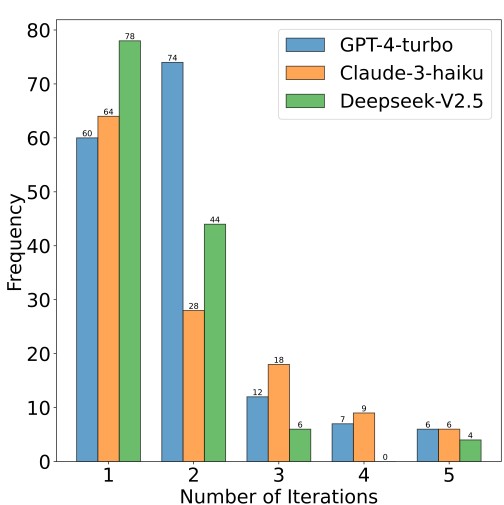

(a) Reaction-level RT validity by molecule with respect to retrieval quality and synthesis difficulty.

(b) No. of iterations it takes for LLMs to generate an RT-valid route.

Figure 4: Details of LLM-generated RT valid retrosynthesis routes with respect to synthesis difficulties or the number of iterations.

in which only representative routes were fed into GPT-4-turbo as reference routes to guide the generation process, and we report the performance in the initial guess, as shown in Table 2. Compared to our current setting, where routes from structurally similar molecules are used as references, this approach fails to capture the dynamics of chemical reactions and struggles to generate round-trip valid reactions or routes (24.42% vs 51.64%). However, the molecule validity remains relatively high (80.6% vs. 87.29%), suggesting that the LLM can understand the syntax of SMILES from those example routes given. An experiment of RAG based on a chiral-aware molecular fingerprint (Orsi & Reymond, 2024) is presented in Table A5.

| Metric / Method | Rouge ↑ | Bleu ↑ | Exact Match ↑ | Molecule Validity ↑ | Route Validity ↑ | No. of Valid Route ↑ |
|---|---|---|---|---|---|---|
| Representative-only | 0.5800 | 0.6051 | 0.00% | 80.65% | 24.42% | 0 |
| Similarity-based | 0.6615 | 0.7048 | 8.00% | 89.63% | 51.64% | 60 |

Table 2: Ablation study of solely using representative routes as references to the LLMs for initial route generation.

**LLMs "Cheat" but Iterative Refinement and Formatter Can Intervene.** LLMs often "cheat" during retrosynthesis route generation when faced with difficult-to-synthesize molecules or complex chemical structures. They may improperly split a SMILES string into two halves, disregarding the chemical validity of the resulting fragments and the reaction itself. Additionally, they might prematurely halt the process, falsely claiming that these challenging molecules are commercially available to avoid further synthesis planning. As shown in Figure 5a, the LLM generates 'COc1ccc(C(=O)OH)c(O)c1O' in its response with an extra hydrogen atom, rendering it an invalid SMILES. The LLM also falsely claims that the molecule 'COc1ccc(C(=O)Cl)c(O)c1O' is commercially available, despite it not being found in any database of purchasable molecules. Another common issue is the erroneous placement of the product, either final or intermediate, back into the reactants, indicative of a faulty or incomplete synthesis route as shown in Figure 5c. With the use of feedback and a formatter, those 'cheated' responses can be corrected as demonstrated in Figure 5b and Figure 5d.

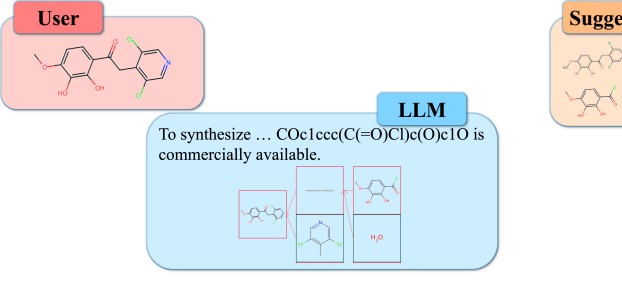

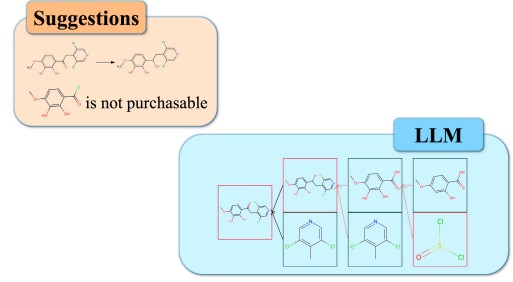

(a) LLM produces molecule against SMILES grammar, claims commercial availability wrongly and stops retrosynthesis prematurely.

(b) With proper feedback, LLM may replace the reaction containing invalid molecules with a round-trip valid reaction.

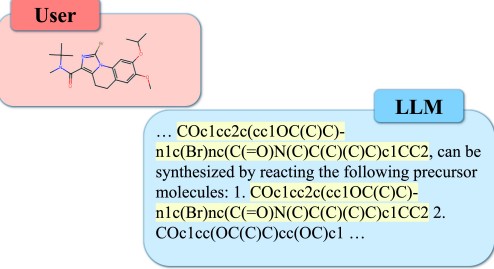

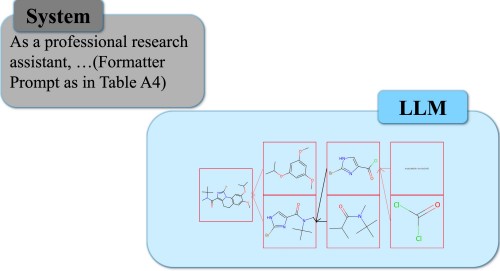

(c) LLM includes the product as a reactant in its predicted retrosynthesis route.

(d) Duplicate products can be identified and removed by the LLM-backed formatter.

Figure 5: LLMs 'cheat' in initial responses but are later corrected within our proposed framework.

**Routes Are Indeed Refined During Iterative Refinement Process.** The iterative refinement process functions more than correcting the formats of LLM-generated routes and addressing errors, it is also the key factor in ensuring the generation of round-trip valid routes. The distribution of the number of iterations required for LLMs to produce a round-trip valid route is shown in Figure 4b. Although LLMs can generate a round-trip valid route using RAG without refinement in some cases, a significant number of valid routes are discovered only after several iterations of refinement. On average, GPT-4-turbo requires 1.90 iterations to generate a round-trip valid route, compared to 1.92 for Claude-3-haiku and 1.55 for Deepseek-V2.5. At the reaction level, the iterative refinement process significantly increases the reaction-level top-5 round-trip validity, as shown in Table 3. For

routes generated with GPT-4-turbo, reaction-level round-trip validity improves from 51.64% in the first iteration to 89.81% in the final iteration.

**Balance Between Prior Chemical Knowledge and General Ability** As shown in Figure 4b, Deepseek generates the highest number of round-trip valid routes in the first iteration, suggesting a stronger capability in handling chemistry-specific tasks. However, the reaction validity of Deepseek-generated routes declines as the process continues as shown in Table 3. Upon manual inspection of the routes across iterations, it was found that the Deepseek-backed formatter may not properly convert routes from textual descriptions to retrosynthesis trees in JSON format, revealing a limitation in its general instruction following capabilities. Ablation experiments using the Deepseek-backed generator and GPT-4-turbo-backed formatter are presented in Table 4. With the GPT-4-turbo-backed formatter, Deepseek shows significantly improved results. Molecule validity increases from 86.76% to 93.45%, and reaction round-trip validity improves from 52.44% to 75.42%, resulting in a 5% increase in overall route round-trip validity. This introduces another design consideration, balancing capabilities between the chemistry domain and the general domain.

| Reacion RT Validity ↑      Method | Retro*(5) | GPT-4-turbo | Claude-Haiku | Deepseek-V2.5 | Deepseek-V2.5 with GPT-4-turbo Formatter |
|---|---|---|---|---|---|
| Before Refinement | 94.34% | 51.64% | 52.89% | 63.67% | 60.36% |
| After Refinement | | 89.81% | 67.18% | 52.44% | 75.42% |

Table 3: Reaction-level top-5 RT validity before and after five rounds of refinements.

| Metric      Method | Query Success Rate↑ | Rouge↑ | Bleu↑ | Exact Match↑ | Molecule Validity↑ | Route Validity↑ | Average Length of Valid Route↓ | Reaction RT Validity of Final Prediction↑ |
|---|---|---|---|---|---|---|---|---|
| Deepseek Generator w Deepseek Formatter | 100.00% | 0.7399 | 0.4428 | 17.00% | 86.76% | 67.50% | 2.75 | 52.44% |
| Deepseek Generator w GPT-4 Formatter | 100.00% | 0.7541 | 0.6742 | 18.50% | 93.45% | 72.50% | 2.85 | 75.42% |

Table 4: The overall performance hinges on the performance of the formatter.

# 5 RELATED WORKS

## 5.1 RETROSYNTHESIS PLANNING

In the early days, from the proposal of retrosynthesis by Corey (1967) in the 1960s to the first decade of this century, chemists relied heavily on their expertise and rule-based methods to navigate the complex space of chemical transformations. This approach, while effective for certain problems, was limited by the scope and scalability of human intuition and manually curated rules. The landscape began to shift with the pioneering work of Segler et al. (2018), who introduced Monte Carlo Tree Search (MCTS) augmented with neural networks. This method represented a significant breakthrough, using machine learning to predict reaction outcomes and guide the search process, marking one of the first instances where computational intelligence illuminated retrosynthetic planning. Building on this foundation, several advanced algorithms have since been developed to explore the vast space of chemical transformations further. Hong et al. (2023) further improves the efficiency and performance of MCTS by replacing rollouts with an experience-guided neural network. Retro* (Chen et al., 2020) and RetroPrimeWang et al. (2021) employ a trained value function to steer the expansion with A* algorithm (Hart et al., 1968). Schwaller et al. (2020) employs a hyper-graph exploration strategy backed by forward reaction prediction models and synthesis difficulty scores. Reinforcement learning-based methods, such as those proposed by Yuan et al. (2024); Yu et al. (2022); Schreck et al. (2019), update the scoring function during self-play, allowing the model to learn and improve its decision-making by simulating retrosynthetic pathways. updates the score function during self-play. Yu et al. (2022) applied a goal-driven actor-critic reinforcement-learning agent to guide the expansion and Yuan et al. (2024) utilized a critic model of route quality based on yields. Aligned with previous efforts in route-aware retrosynthesis planning, Liu et al. (2023b) represented the current synthetic route as graphs to facilitate single-step retrosynthesis predictions.

Despite significant advancements in machine learning-based retrosynthesis planning, leading commercial software like SYNTHIA™ (formerly Chematica (Grzybowski et al., 2018)) continues to

integrate expert-encoded chemical rules with sophisticated algorithms. This approach underscores the importance of combining human expertise with data-driven models to enhance the accuracy and reliability of synthetic pathway predictions. As highlighted by Strieth-Kalthoff et al. (2024), the future of retrosynthesis planning lies in the tight cooperation between data-intensive models and human expertise, ensuring that computational predictions are both feasible and scalable.

## 5.2 LLM FOR PLANNING

Recent advances in large language models (LLMs) have revolutionized the way complex planning processes are approached on diverse tasks from internet browsing to robotics (Hu et al., 2024; Prasad et al., 2024; Sun et al., 2023; Zhang & Lu, 2024; Tan et al., 2024; Zhou et al., 2023; Yu et al., 2024a; Shinn et al., 2023; Yao et al., 2023; Lu et al., 2023; Trinh et al., 2024).

One diversion between LLMs and traditional planners is that LLMs may produce plans that include impossible actions. To ground LLMs to the feasible action spaces, previous approaches provide available actions in different granularity as external tools to agent-like planners (Prasad et al., 2024; Hu et al., 2024) or as functions to code-style planner Sun et al. (2023); Zhang & Lu (2024). Some approaches (Liu et al., 2023a) also use LLMs as a format converter to translate actual problem text into sequences compatible with traditional planners and use them to solve the planning problem.

Another important designing factor in LLM planning is the utilization of feedback provided by the underlying systems. Methods like Chain-of-thought (Wei et al., 2023), Least-to-most (Zhou et al., 2023) generate plans in a single pass without taking any feedback. ReAct (Yao et al., 2023) decouples one planning step into a dedicated reasoning stage and an acting stage, where observations in previous steps are utilized in the reasoning stage of the current step. Adaplanner (Sun et al., 2023) further refines the entire plan with feedback on the execution results of the current plan from LLMs.

Albeit the rapid development of LLMs in the field of planning, Kambhampati et al. (2024) advocated that LLMs themselves cannot perform planning due to the underlying n-gram-like generation mechanism. Instead, Kambhampati et al. (2024) suggested that LLMs can be utilized in the overall planning process as auxiliary parts like format translators, summarizers whereas the soundness of the entire planning pipeline is bounded by the experts (human beings or models) involved. Our method aligns with this philosophy where the local experts back the soundness of generated retrosynthesis routes, and also with the philosophy suggested by Strieth-Kalthoff et al. (2024).

## 6 CONCLUSION & FUTURE DIRECTIONS

In this work, we have introduced a new methodology for retrosynthesis route generation and proposed a novel framework for leveraging LLMs with RAG and iterative refinement through expert feedback. Our approach demonstrates the ability of LLMs to successfully generate retrosynthesis routes with high query success rates and competitive route quality compared to traditional methods. Notably, the iterative refinement process enhances the feasibility of generated routes, addressing the challenges associated with round-trip validity and retrieval quality.

Despite promising results, challenges remain in balancing LLM generalization with domain-specific chemical knowledge. Expanding the route database and enhancing feedback mechanisms could improve performance and enable advanced expert systems with human-in-the-loop strategies. Our experiments also indicate that instruction-based fine-tuning struggles to fully capture retrosynthesis complexities, highlighting the need for improved tokenization or training objectives. Additionally, the scalability of our approach is limited by latency of token generation. While we avoided sampling for reproducibility, future work could explore sampling multiple routes in a single pass to address this limitation.

Overall, our findings suggest that LLMs hold significant potential for automating complex retrosynthesis tasks, paving the way for more efficient and scalable approaches in chemical synthesis planning. Future directions will explore the integration of dynamic knowledge updates and optimization of LLM prompt structures to further enhance performance in out-of-distribution scenarios.

## REPRODUCIBILITY STATEMENT

Weights for all models are downloaded from their official repositories respectively[5]. We set LLMs' sampling temperature to 0 during route generation for reproducibility and set the temperature to 0.2 during formatting.

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

# A APPENDIX

## A.1 AND-OR TREE ABSTRACTION USED IN PRIOR APPROACHES

Previous retrosynthesis planning approaches abstract the problem as navigating through an AND-OR tree as shown in Figure A1.

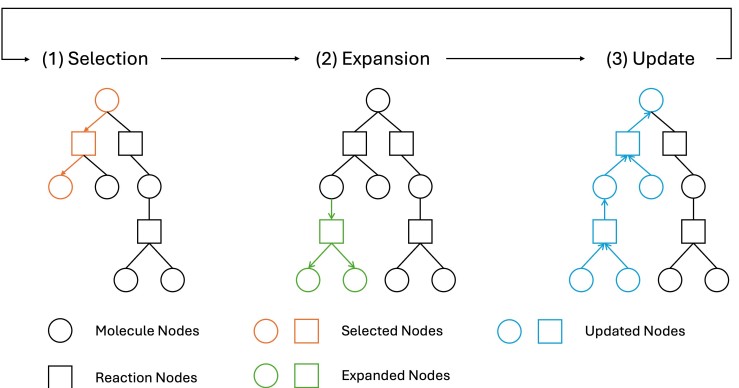

Figure A1: The traditional paradigm of planning-based retrosynthesis route generation represents the process as an AND-OR tree, where molecule nodes serve as OR nodes and reaction nodes as AND nodes. First, the planner selects a node to expand based on estimated cost. Next, a single-step retrosynthesis reaction is applied to the molecule nodes, expanding them into an AND-OR subtree. Finally, the cost along the pathway is updated in preparation for the next selection. The AND-OR tree is rooted on the target molecule node and has commercially purchasable materials as leaf nodes.

## A.2 DATASET STATISTICS

We report the dataset statistics in Table A1. The test subset consists of the first 200 routes from the test set and is slightly more challenging to synthesize compared to the overall test set.

| Split | No. of Routes | Avg. Route Length | Avg. SA Score |
|---|---|---|---|
| Train | 299202 | 1.76 | 2.64 |
| Test (all) | 79529 | 3.04 | 2.77 |
| Test (subset) | 200 | 3.12 | 2.81 |

Table A1: Statistics of the dataset used in the experiments. SA (synthetic accessibility) Score (Ertl & Schuffenhauer, 2009) is a heuristic metric to evaluate the difficulty of synthesis, with 10 being the hardest and 1 being the easiest.

## A.3 SUGGESTION AND FEEDBACK SCHEMA

We summarize various scenarios where the LLM may generate incorrect routes, along with corresponding descriptions and suggestions for refinement, as shown in Table A2.

## A.4 PROMPTS USED

We provide the prompts used in our experiments in Table A3 and Table A4. Placeholders enclosed by {{}} are used and will be filled with corresponding text during inference. While we manually refined our prompts, there remains room for improvement, and further refinement could enhance overall performance. We leave tasks such as automatic prompt construction for future work.

| Scenario | Description | Suggestion |
|---|---|---|
| Wrong Product | The final product molecule is not the target molecule. | Round-trip valid single-step retrosynthesis reactions for the product molecule. |
| Cyclic Route | Duplicate product molecules appear in the reactants. | NA |
| Disconnected Route | Invalid JSON format. | NA |
| Unnecessary Subroute | Molecule {mol smiles} is already available. | You must stop its synthesis here. |
| Not Round-trip Valid | Reaction {reaction smiles} is not round-trip valid. | Round-trip valid single-step retrosynthesis reactions for the product molecule. |
| | | I cannot find any valid reactions to replace it. Please restart from its precursor. |
| Invalid Product Molecule | Product molecule {mol smiles} in reaction {reaction smiles} is invalid. | NA |
| Invalid Reactant Molecule | Reactants molecule(s) {mol smiles} in reaction {reaction smiles} is invalid. | Round-trip valid single-step retrosynthesis reactions for the product molecule. |
| Duplicate Product in Reactants | Product molecule {mol smiles} appears in the reactants. | NA |
| Unavailable Starting Material | Starting material molecule {mol smiles} is not commercially available. | Round-trip valid single-step retrosynthesis reactions for the product molecule. |
| | | I cannot find any valid reactions to replace it. Please restart from its precursor. |

Table A2: Feedback templates for different scenarios are provided. The text in the "description" and "suggestion" columns will be fed to the LLM to assist in refining the route. Placeholders {} are used, with specific molecules or reaction SMILES provided in the actual suggestions.

---

**System Prompt for Route Generation**

---

As a professional chemist specialized in synthesis analysis, you are tasked with generating a full retrosynthesis route for a target molecule provided in SMILES format.
A retrosynthesis route is a series of retrosynthesis reactions that starts from the target molecule and ends with some commercially purchasable compounds.

To assist you, example retrosynthesis routes that are either close to the target molecule or representative will be provided.
Your responses will receive reaction-wise feedback from chemical experts based on the validity of SMILES inside each reaction and the round-trip validity of the reaction.
A round-trip valid reaction means that it's predicted to produce the products in the reaction given the reactants.

You will start with the provided examples and iteratively improve your retrosynthesis route based on the feedback.

Here's the Step-by-Step Breakdown for this task:

Step 1: Identify the target molecule.
Step 2: Decompose the target molecule into precursor molecules. Note that precursor molecules are not necessarily smaller and simpler to synthesize, you should prioritize those molecules that you already know how to synthesize,even though they might be harder to synthesize.
Step 3: For each precursor molecule, repeat the decomposition process until the simplest starting materials are reached.

The decomposition process should be recursive, where each precursor can further break down into its own precursors.

Here is the format of target molecule provided:
<target_molecule>
{{
TARGET_MOLECULE
}}
</target_molecule>

Here is the format of example retrosynthesis routes:
<example_routes>
{{
EXAMPLE_ROUTES
}}
</example_routes>

After your initial attempt, you will receive feedback in the following format:
<feedback>
{{
FEEDBACK
}}
</feedback>

Based on this feedback, generate an improved retrosynthesis route, again following the same format as shown in the examples.
Only present the retrosynthesis route and nothing else.

---

Table A3: System Prompt for Retrosynthesis Route Generation with Refinements

**Retrieved Routes Templates**

{
"TargetMolecule":
{{example_mol}}
"RetrosynthesisRoute":
{{example_route}}
}

**JSON Formatter Prompt Template and Template for Retrieved Routes**

Instruction
———

As a professional research assistant, it is your job to convert the text description of a retrosynthesis route into a tree in JSON format.
The given description may not be complete, it is also your job to complete the reactions in the route with the correct retrosynthesis reactions.
No reagent information is required, please exclude them from your responses.
In the case that IUPAC names appear in the description, use provided tool to convert it to a SMILES string.
Output format
———

Your output should be in JSON format like
{
  "molecule": "TARGET_MOLECULE_SMILES",
  "children": [
    {
      "molecule": "PRECURSOR_1_SMILES",
     "children":
        ...
     ]
    },
    {
      "molecule": "PRECURSOR_2_SMILES",
     "children": [
      {
        "molecule": "SUB_PRECURSOR_1_SMILES",
       "children": [
         ...
       ]
      }
     ]
    }
  ]
}
where key "molecule" is the synthesis result of all molecules in the key "children", in a recursive manner.
Key "molecule" must be a valid SMILES representation of a valid molecule, you can leave "children" as an empty list if it's commercially available.
User's input
———

Input route description:
{{input}}

Table A4: Prompt Template for JSON Formatter

## A.5 Pseudocode of Proposed Approach

---

**Algorithm 1:** Retrosynthesis Route Generation with LLM

---

**Input:** Target Molecule $t$; Iteration Budget $N$; Frozen Pretrained LLM with Weights $\theta$;
**Output:** Predicted Retrosynthesis Route $R$

(1) `GenerateRoute`($t$, $N$)
(2) $i \leftarrow 0$;
(3) **while** $i \geq N$ **do**
    /* Molecular-Similarity-Based RAG          */
(4)  **if** $i = 0$ **then**
(5)    $\{R_{sim}^T\} \leftarrow Retrieve(DB_{Route}, t)$;
(6)    $\{R_{sim}^J\} \leftarrow Describe(\{R_{sim}^J\})$;
(7)    $R_i^T \leftarrow f(template\_gen(t, \{R_{sim}^T\})|\theta)$;
(8)  **else**
(9)    $R_i^T \leftarrow f(template\_gen(t, \{R_{sim}^T\}, R_{i-1}^T, suggestion_{i-1}|\theta)$ ;
(10)  **end**
    /* LLM-backed Formatter                         */
(11)  $R_i^J \leftarrow f(template_{format}(R_i^T)|\theta)$;
    /* Suggestion from Expert Models      */
(12)  $suggestion_i, frontier_i \leftarrow suggest(R_i^J)$;
    /* Local Knowledge Base Update        */
(13)  **if** $frontier_i \equiv \varnothing$ **then**
    // $R_i^J$ is Valid
(14)    Update $DB_{Route}$ with $R_i^J$;
(15)    **break**;
(16)  **else**
(17)    **if** $t \notin frontier_i$ **then**
        // $R_i^J$ is Partially Valid
(18)      $sub\_traj \leftarrow \{\}$;
(19)      **for** *each new target $t'$ in $frontier_i$* **do**
(20)        **if** $t' \in DB_{subroutes}$ **then**
(21)          $sub\_traj[t'] \leftarrow DB_{subroutes}[t']$
(22)        **else**
(23)          $sub\_traj[t'] \leftarrow$ `GenerateRoute`($t', \max(N-1, 0)$);
(24)          $DB_{subroutes}[t'] \leftarrow sub\_traj[t']$
(25)        **end**
(26)      **end**
(27)      $R_i^J \leftarrow Insert(R_i^J, sub\_traj)$;
(28)      $suggestion_i \leftarrow suggest(R_i^J)$;
(29)    **end**
(30)  **end**
(31)  $i \leftarrow i + 1$;
(32) **end**
(33) **Return** $R_i^J$;

---

## A.6 Case Analysis

We present three case studies: where the LLM generates a round-trip valid route while Retro* fails to (1) generate a valid route or (2) even provide a prediction, and (3) where Retro* generates a round-trip valid route while the LLM fails to generate a valid route, as illustrated in Figure A2.

In the first case, Retro* may fail to generate a round-trip valid route due to an invalid reaction in one step of the retrosynthesis process, as shown in Figure A2a. In contrast, the LLM corrects its initial prediction in the first trial and successfully generates a fully valid round-trip route.

It is also possible for Retro* to exhaust its entire iteration budget (500 iterations), as demonstrated in Figure A2b. In this case, Retro* spends excessive resources on subroutes that lead to dead ends, such as 'ClP(Cl)(Cl)(Cl)Cl' and 'O=S(Cl)Cl', for which no expert can provide any valid single-step retrosynthesis suggestion. However, the LLM overcomes this challenge by leveraging expert advice to generate a valid route. Upon manually examining the suggestions that contributed to the valid route, we found that all of them came from the same single retrosynthesis model used by Retro*. This suggests that the LLM can effectively select a single-step model leading to overall success, even without relying on an explicit value function.

The LLM may also encounter dead ends, as shown in Figure A2c, where several attempts are made to synthesize 'CCOC(=O)c1c(O)c2cc(Br)c(C)c(C)c2oc1=O'. However, it is worth noting that, compared to the 500 iteration budget used by Retro*, we only utilized 5 iterations to refine the route. With additional iterations, the issue may potentially be resolved.

## A.7  USER INTERFACE FOR THE RETROSYNTHESIS AGENT

We showcase the user interface of our proposed framework, hosted on a local server, as illustrated in Figure A3. The user inputs the SMILES representation of the target molecule, and RAG is performed implicitly. Once the LLM generates a route prediction, the user can either provide feedback directly or opt for 'surrogate' feedback generated by local expert models. This process continues until a configurable iteration budget is reached or a valid route is found.

## A.8  USING 3D-AWARE MOLECULAR FINGERPRINTS FOR RETRIEVAL

We rebuilt our route database using MinHashed Atom-Pair for Chiral (MAP4C), a chiral-aware molecular fingerprint described by Orsi & Reymond (2024) and evaluated its impact on Retrieval-Augmented Generation (RAG) performance. Using DeepSeek, we tested this approach on the Retro* dataset after a single iteration, presenting empirical results highlighting molecular fingerprints' influence on RAG efficacy.

| Metric
Retrieval Scheme | Rouge | Bleu | Exact Match | Molecule Validity ↑ | Reaction RT Validity ↑ | No. of RT Valid Routes↑ |
|---|---|---|---|---|---|---|
| No RAG at all | 0.4485 | 0.2900 | 0.00% | 58.31% | 30.00% | 1 |
| Morgan Fingerprints | 0.6605 | 0.5611 | 9.14% | **87.68%** | **46.20%** | **78** |
| MAP4C Fingerprints | 0.6442 | 0.5921 | 7.54% | 89.55% | 43.21% | 36 |

Table A5: Ablation study of using chiral-aware molecular fingerprints for similarity calculation in retrieval.

## A.9  EVALUATION ON CHEMISTRY-AWARE LLMS

We evaluated ChemDFM-v1.5-8B (Zhao et al., 2024) for retrosynthesis route generation under four configurations: (1) Vanilla, (2) fine-tuned, (3) RAG-enhanced, and (4) fine-tuned with RAG. Fine-tuning was performed on the Retro* training dataset to generate JSON-formatted routes from product SMILES, using low-rank adapters to reduce trainable parameters. In RAG settings, reference routes were extracted as described in the main text and provided only during inference. We report the evaluation results in Table A7. It is worth noting that while fine-tuned LLMs achieve similar results in text-related metrics like BLEU and Rouge, meaning fine-tuned LLMs possess the ability to produce retrosynthesis trees similar to the ground truth as provided for supervised fine-tuning, they struggle to generate valid routes, even pretrained with chemistry knowledge. Compared to our approach without refinement,

Table A6: Details of supervised fine-tuning baseline.

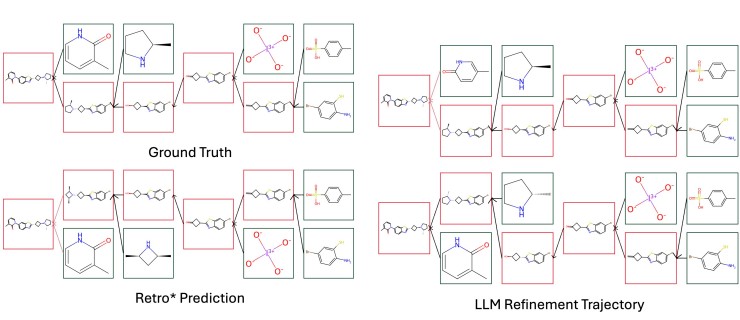

(a) LLM gives a round-trip valid route while retro* cannot.

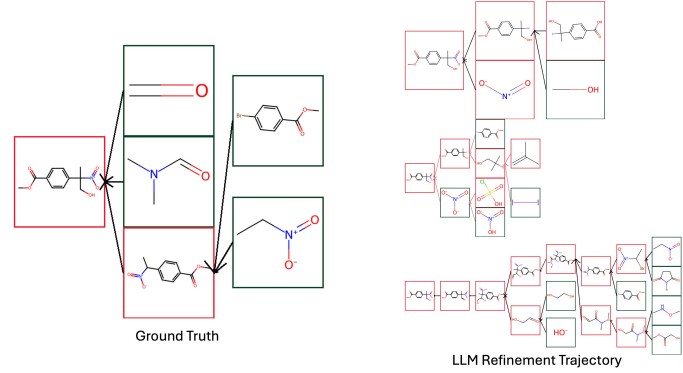

(b) LLM gives a round-trip valid route while retro* failed to find one route.

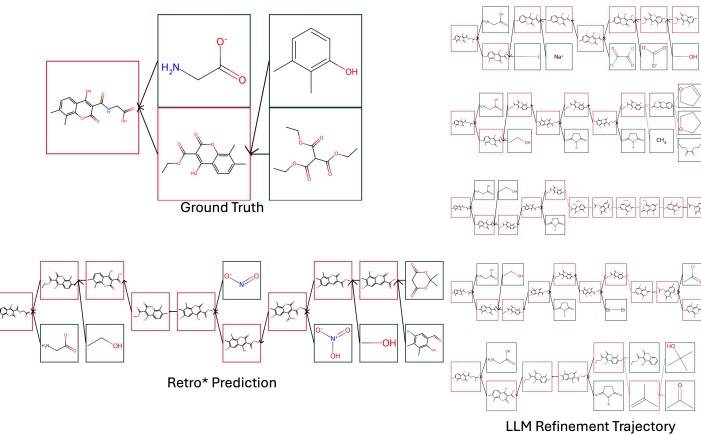

(c) Retro* gives a round-trip valid route while LLM cannot.

Figure A2: Three case analyses were visualized. Molecules in green boxes are commercially available, while those in red boxes are not. Reactions in black indicate round-trip validity, whereas those in red do not.

## 🔗 Retrosynthesis Agent

🤖 Hi, I'm a chatbot who can generate retrosynthesis routes given a product molecule SMILES. How can I help you?

🔴 COC(=O)c1cccn1CCF

🤖 {"molecule": "COC(=O)c1cccn1CCF", "children": [{"molecule": "FCCl", "children": []}, {"molecule": "COC(=O)c1ccc[nH]1", "children": []}]}

🔴 surrogate

🔴 <reaction feedback> Reaction COC(=O)c1cccn1CCF>>FCCl.COC(=O)c1ccc[nH]1 is top-5 round-trip valid and all SMILES inside are valid. This reaction should be valid in practice. </reaction feedback>

Figure A3: Demonstration of Web UI

| Metric
Model | Query
Success Rate↑ | Rouge↑ | BLEU↑ | Exact Match↑ | Molecule Validity ↑ | Reaction RT Validity ↑ | Route Validity↑ | Average Length
of Valid Route ↓ |
|---|---|---|---|---|---|---|---|---|
| Vanilla | 8.50% | 0.3545 | 0.4771 | 0.00% | 0.00% | 0.00% | 0.00% | NA |
| Vanilla w RAG | 5.50% | 0.3805 | 0.4738 | 0.00% | 0.00% | 0.00% | 0.00% | NA |
| Finetuned | 100.00% | 0.5513 | 0.6948 | 10.00% | 98.28% | 73.50% | 26.00% | 1.87 |
| Finetuned w RAG | 100.00% | 0.6692 | 0.6724 | 9.50% | 98.48% | 73.83% | 26.50% | 1.81 |

Table A7: Performance of local finetuned LLMs on retro* dataset.

## A.10 EVALUATION ON PISTACHIO DATASET

We extract synthesis routes from the Pistachio dataset after applying rule-based filtering to remove reactions that are either incomplete or uninformative. Following the approach of Chen et al. (2020), we construct the training dataset of routes using reactions from the training set, the testing dataset of routes using reactions from both the training and testing sets, and the validation dataset of routes using reactions from both the training and validation sets. We further remove duplicate routes in testing or validation datasets that appear in the training set, and filter out routes that are directly synthesised by starting materials. We assess our model on 50 routes in the testing dataset. The statistical details of the Pistachio dataset are provided in Table A8. One thing to note that our filtering includes removing non-necessary reactants from the reactions (reactants with molecular similarity compared to the product less than a threshold) and also breaking multi-products reaction into several reactions, together with the distribution shifts from USPTO to Pistachio, expert models may not perform well on the new route dataset. We present the results on the Pistachio dataset in Table A9. Our proposed method, using Deepseek, achieves a higher ROUGE score but a lower BLEU score. Upon careful examination, the low BLEU score is attributed to hallucinations by the LLM, which generate unwanted lengthy sequences for certain molecules. Additionally, our method achieves a higher exact match rate, primarily due to the overlapping reactions used in constructing different dataset splits. Lastly, we observe a comparable route validity between our approach and EG-MCTS.

| Split | No. of Routes | Avg. Route Length | Avg. SA Score |
|---|---|---|---|
| Train | 73256 | 2.96 | 2.91 |
| Validation | 343 | 2.95 | 3.22 |
| Test (all) | 308 | 3.00 | 3.19 |
| Test (subset) | 50 | 2.76 | 3.18 |

Table A8: Statistics of Pistachio dataset used in the experiments. SA (synthetic accessibility) Score (Ertl & Schuffenhauer, 2009) is a heuristic metric to evaluate the difficulty of synthesis, with 10 being the hardest and 1 being the easiest.

| Model / Metric | Query Success Rate↑ | Rouge↑ | BLEU↑ | Exact Match↑ | Molecule Validity ↑ | Route Validity↑ | Average Length of Valid Route ↓ |
|---|---|---|---|---|---|---|---|
| Retro* | 94.00% | 0.7177 | 0.5281 | 6.00% | 100.0% | 72.00% | 3.28 |
| EG-MCTS | 92.00% | 0.6985 | 0.5285 | 6.00% | 100.0% | 56.00% | 1.96 |
| Deepseek | 100.00% | 0.7479 | 0.2682 | 46.00% | 92.59% | 60.00% | 2.57 |

Table A9: Performance on pistachio route dataset.

