# OpenReview forum: "How Well Can LLMs Synthesize Molecules? An LLM-Powered Framework for Multi-Step Retrosynthesis"
_ICLR.cc/2025/Conference — ICLR 2025 Conference Withdrawn Submission_

### Official Review · Reviewer_BRau · 2024-10-23

**Soundness:** 3
**Presentation:** 2
**Contribution:** 3
**Rating:** 5
**Confidence:** 3

**Summary:**

The authors present a framework for automated retrosynthesis using large language models. Contrary to the commonly employed strategy of obtaining a synthesis plan by iteratively expanding a tree of reactions and reactants, this approach takes a more holistic view, generating and iterating over the entire sequence as a whole. In the proposed framework, the user inputs a target molecule as query to the retrosynthesis planner. The system then starts by performing a fingerprint-based similarity search in a database of known reaction plans to use the synthesis route of a similar molecule as starting point. This example is fed to the LLM which generates a route for the query molecule, and format it in a structured JSON file. This response then goes into an interative loop which (1) checks the validity and commercial availability of reactions and building blocks (using expert models and other databases) and in case of invalid steps, feeds-back the route and the validity check results to the LLM for further refinement. The authors experiment with three different LLM models (GPT4-Turbo, Claude-Haiku and Deepseek) and compare against Retro*, a well-known retrosynthesis model. They show that the proposed framework effectively combines all of these components and allow to propose valid synthesis routes for about 80% of the tested query molecules.

**Strengths:**

Overall, I found the paper of good quality and very well written. I think it addresses an important challenge (automatic retrosynthesis) from a modern perspective (in combination with LLM and integrating workflows and specialised tools / user feedback). In particular:

- The Introduction and Preliminary sections are well written and effectively guide the reader into the subject matter.
- The presentation of the proposed framework is also very clear (Figure 2).
- I believe that the integration of both specialized models and human feedback (Section 4.1) is a flexible and powerful combination. In particular, Table A2 provides a clear and helpful list of scenarios which could be handled by the framework.
- The experiments section is well structured and clearly support the claims made earlier in the paper.
- The authors clearly identify drawbacks of the proposed methods (lines 333-343, 371-377) showing scientific integrity and motivating future work.

**Weaknesses:**

#### Main weaknesses:
1. My main comment is on the motivation of this line of work. I believe the authors could present more clearly the potential benefits of an eventual transition from traditional synthesis planners to LLM-powered planners in a more effective way. While I have some intuitions about what potential gains in accessibility and ease-of-use such approaches could yield, after reading the paper, I am still left somewhat unclear about which precise drawbacks from traditional planners LLM-based planners aim to address.

2.  The validity-availability iterative loop (steps 3-7 in Figure 2) should be explained in more details before the experimental section. For example, on line 199 it is mentioned that "The synthesis tree is evaluated by a series of expert models for feedback" but no concrete example of such models is given. Although more details are given in pages 5-6, adding more details about expert models in the methods section would make that part better standalone and enhance clarity.

3. The availability of the code to reproduce experiments is not discussed.

4. The quality of several figures could be enhanced -- e.g. very thinly rendered molecular bonds, drafty layout (table 3) etc. In Figures 1 and 5, some elements in the "LLM" box are unreadable. The pseudocode presented in appendix is hard to read (too many different text styles).

#### Minor comments:

1. Broken link in footnote of page 3 ("s" is not in the \url).
2. Figure 4 is missing the caption.

**Questions:**

1. On lines 270-272, the authors describe how predictive models are used for RT assessment on the proposed synthesis plan. Why can't the framework simply employ traditional tools like rdkit to run the proposed reaction and reactant? (why employing ML models here?)
2. I am not sure to understand exactly what is meant by Query Success Rate (line 252). How exactly is that evaluated and how can the model fail to produce any sort of synthesis path when the quality is not taken into account? I believe this could be made clearer with an example.
3. How reliant is the framework on the success of the similarity-based search? Can the model successfully propose synthesis routes for molecules that are synthesizable, but far outside the known synthesis routes database? Results are presented for similar v.s. representative starting points in Table 3, but how about without providing any starting point (just the query molecule)?
4. Regarding scalability with the number of queries: synthesis plans can be designed by hand by chemists on a case-by-case basis. On the automated side, one of the most important applications of automated synthesis planning is to feed high throughput screening (HTS) platforms with hundreds or thousands of molecules that are guaranteed to be synthesizable in a timely fashion (without human involvement). Where do LLM-based planners fit on this scale? Apart from operational costs listed in lines 290-293 (which were helpful), it would be interesting to discuss these limitations and the exact use-cases for the presented framework.
5. Have any experiments been carried out using user feedback in the refinement loop? User feedback is mentioned as a possibility for this framework in the text but no explicit experiments are presented.

---

> ### Author Response · Authors · 2024-11-25
>
> Thank you for your invaluable feedback on our paper. We have addressed the points raised in your review and marked the corresponding sections in red, including: the reproducibility statement, a consolidated and detailed introduction of the iterative refinement loop (with additional details provided in Appendix Table A2), improvements to table quality, enhanced readability of the pseudocode, and an updated caption for Figure 4.
>
> 1. Regarding question 1 (why ML models for RT assessment):
> As far as we know, RDKit is not capable of predicting reactants directly from products. While it can simulate reactions using products to produce potential reactants, this is fundamentally different. Thus, we leverage ML models, which are well-suited for this task.
> 2. Regarding question 2 (query success rate):
> this metric is calculated as the percentage of properly formatted responses. It is valuable because it measures how often our approach can provide a viable solution. This becomes particularly useful for challenging molecules where human experts might turn to our system for creative insights during synthesis planning.
> 3. Regarding question 3 (Robustness without any RAG): We conduct experiment of just providing query smiles for generation in **Appendix Table A5,** and also on finetuned LLM in Table A7, showing much worst performance compared to molecular similarity based RAG.
> 4. Regarding Question 4 (Cost and scalability of LLMs):
> While the operational and computational costs of LLMs are indeed high, they can feasibly sample multiple routes during generation, which enhances scalability. We acknowledge this as a limitation in our conclusion part and aim to explore it in future work.
> 5. Regarding Question 5 (User-involved feedback):
> Currently no experiment with user-involved feedback is conducted. Although no user-involved feedback has been implemented in this work, our experiments demonstrate that LLMs can refine synthesis routes when provided with expert feedback. We believe this capability extends to aligning with human preferences, much as it already aligns with chemical feasibility guided by expert models.
>
> Regarding question 3, LLMs can produce more chemically feasible responses with more similar reference examples as shown in figure 4a. We also performed experiments in **Appendix A.8** with just target smiles provided to the LLM, showing much worst performance in in terms of similarities and also chemically feasibilities.
>
> Regarding question 4, sadly the LLM is expensive both in time and operation costs, but it is feasible to sample multiple routes during generation, which improves scalability. We leave this for future works.
>
> Regarding question 5, currently no user-involved feedback is conducted. But our experiments demonstrated that LLMs can indeed refine the routes given feedback from experts, we believe they will conform human preferences just well as they conform chemical feasibility by the expert models used.
>
> We sincerely appreciate your constructive suggestions and hope that our responses adequately address your concerns.

---

> ### Comment · Reviewer_BRau · 2024-12-01
>
> I wish to thank the reviewer for their thoughtful response and the provided revision of the manuscript. I have carefully read the author response and the revised manuscript. Some of my concerns were answered by the reviewer response:
> - Comment #2 about the validity loop
> - Question 2 on query success rate
> - Question 4 on scalability
>
> However I still have some unresolved concerns:
>
> ### 1. About ML models for RT assessment.
>
> The author's answer was:
>
> > Regarding question 1 (why ML models for RT assessment): As far as we know, RDKit is not capable of predicting reactants directly from products.
>
> That is the backward direction. However it is not clear to me why, given a complete synthetic route, this route could not be validated by simulating each reaction using RDKit in the forward direction?
>
> ### 2. About the availability of the code
>
> The Reproducibility Statement added to the revised manuscript does not address the availability of the code used to reproduce the experiments in the manuscript, which I explicitly enquired about, but only the availability of the pre-trained models used as part of this system.
>
> ### 3. About the quality & readability of the figures
>
> I do not find that the quality and readability of the figures and pseudocode has been noticeably improved in this revision.
>
> ### 4. Regarding the dependency on RAG data
>
> The authors responded with added experiments without RAG data. I believe these experiments add important context and are greatly appreciated. At the same time, the performance greatly decreased of the system without that data represents an important limitation of the system for its useful deployment on new chemical entities with no close analogs in existing synthesis databases. I believe this limitation should be addressed or at least discussed in the manuscript.
>
> ### 5. My first comment listed under "Weaknesses" does not seem to have been addressed in the revision:
>
> My main comment is on the motivation of this line of work. I believe the authors could present more clearly the potential benefits of an eventual transition from traditional synthesis planners to LLM-powered planners in a more effective way. While I have some intuitions about what potential gains in accessibility and ease-of-use such approaches could yield, after reading the paper, I am still left somewhat unclear about which precise drawbacks from traditional planners LLM-based planners aim to address.
>
> ### 6. My comment about the mention of expert feedback
>
> I believe the capability to integrate expert feedback is one of the strengths of this paradigm. Unfortunately, since as the authors mention, `"no experiment with user-involved feedback is conducted"`, I believe it constitutes an unsupported claim to state that the model supports such feedback. While it certainly opens the way to such user-based feedback, I believe the wording surrounding such feedback is still misleading the reader in some areas of the paper. For exemple in the abstract (line 18) `"to generate an initial retrosynthesis route, which is then iteratively refined through expert feedback"` and on line 138 `"an expert-powered feedback module"`. If these occurrences strictly refer to expert *models*, as it seems to be, this should be clarified (no human experts). If they also include human experts, then experiments should support this claim. Note that the absence of human expert feedback experiments has been made clear in other parts of the paper, e.g. in the added paragraph, on lines 233-236,. However the set of claims and the impressions made on the reader need to be consistent throughout the entire paper (including in the abstract and introduction).
>
> #### Typos:
>
> - On line 135, it would make more sense to label the final route $R_T$ rather than $R_t$
>
> ---
>
> Overall, I find that the revision improved the paper. However, these improvements fall short of what I expected, and my initial rating of 6 was somewhat conditional to these ameliorations, which mostly consisted of clarifications and were reasonably within the reach of a rebuttal period. For these reasons I must for the moment revise my score to 5: slightly below acceptance threshold. I believe this paper has merits and should eventually be published, but does not at the present moment meet the standards of the conference.

---

### Official Review · Reviewer_daJj · 2024-10-28

**Soundness:** 3
**Presentation:** 3
**Contribution:** 2
**Rating:** 3
**Confidence:** 3

**Summary:**

This paper addresses the challenge of computer-aided synthesis planning (CASP), i.e. finding viable synthetic routes to a given target molecule. For this purpose, the authors introduce a two-step LLM-centric pipeline: First, a full route is proposed through retrieval-augmented generation from the LLM. In the second step, this route is refined using feedback from multiple specialist models. This method is benchmarked against a “classical” tree search approach using a neural-network-enhanced A* algorithm. The paper also discusses empirical insights gained from model evaluation.

**Strengths:**

-	Most current CASP methods rely on iteratively navigating the retrosynthesis tree, which struggles with with strategic multi-step planning. Against this background, the idea of generating an initial complete route, followed by route refinement, is promising.
-	Empirical results indicate that retrieval-augmented generation enhances the quality of initial route proposals.
-	The paper is generally well-written, and its logic is easy to follow.

**Weaknesses:**

-	Benchmark results do not demonstrate a performance advantage over the baseline strategy – even given that the problems are relatively simple (see below). Therefore, application to real-world problems is still distant. A broader evaluation, potentially including MCTS-based methods, might yield more comprehensive insights.
-	The significance of the benchmark set is doubtful for two reasons: 1) The problems are relatively simple – average route lengths of <5 steps do usually not require advanced, multi-step strategic thinking. 2) The problems are extracted from the patent literature, which faces well-known data quality issues. For example, a rapid visual inspection of the “ground truth” routes in Figure A.3 reveals synthesis-relevant chemical errors in two out of three examples, further questioning the reliability of these benchmarks.
-	The discussion of literature precedence, particularly regarding computer-aided synthesis planning, is incomplete, and omits a number of pioneering contributions: The seminal retrosynthesis works by Corey; the first demonstrations of retrosynthesis with ML / AI (Segler, Coley), and the state-of-the-art in CASP (Grzybowski) – which is, as of today, not based on AI technology.

**Questions:**

-	The discussion of “expert models” in the main text implies that they provide ground-truth information. However, the reaction-wise feedback models are subject to severe limitations themselves, especially due to the data they are trained on. This should be discussed in the main text.
-	The RAG model retrieves reference syntheses based on a hard-coded notion of “chemical similarity” (fingerprint similarity), assuming that similar molecules exhibit similar synthetic routes. However, there are many examples that violate this assumption, especially when stereochemistry comes into play. At minimum, this limitation should be acknowledged. In the long term, the RAG should be based on similarity measures which are more meaningful to the similarity of synthetic routes.
-	Is the “query success rate” a meaningful metric, especially in comparison to the non-LLM-based baseline model?
-	Chapters 3 and 4.1 have overlapping content. A single, consolidated section that discusses the experimental setup, particularly with respect to expert models and knowledge databases, would improve readability.
-	The term “round-trip validity” is confusingly applied. The original definition from Schwaller et al. only defines it for single reaction steps (target molecule –(retro-direction model)–> starting material(s) –(forward-direction model)–> target molecule). The authors later use the term for scoring full synthetic routes (but without defining what a “round-trip valid route” is). This should be clarified. Moreover, the limitations of their method to determine round-trip validity (p.5) should acknowledged.

---

> ### Author Response · Authors · 2024-11-25
>
> Thank you for your valuable feedback on our paper. We have marked in orange the content addressing your comments, including: the discussion of expert model limitations (Question 1), a consolidated explanation of expert models and knowledge bases (Question 4), clarification of route validity and reaction round-trip validity (Question 5), the addition of the EG-MCTS baseline (Weakness 1), an expanded literature review on retrosynthesis planning (Weakness 3), new experiments incorporating chiral-aware fingerprints (Question 2), and additional testing using the Pistachio dataset (Weakness 2).
>
> Regarding question 2 about using route similarity instead of molecular similarity:
> Our algorithm operates with only the target molecule as input, making route-level comparisons infeasible at the outset since synthesis routes are not known at that stage. However, we do leverage route-level similarity in the clustering of our database, enabling us to provide reference routes for out-of-distribution molecules. This is further detailed in Section 3 (Molecular-Similarity-Based RAG).
>
> Regarding question 3 about query success rate: Query success rate is an important metric as it measures the frequency with which our approach generates solutions that are properly formatted. This is particularly valuable for hard-to-synthesize molecules, where human experts might rely on our system for creative insights during synthesis planning while traditional planners even failed to provide a non-perfect starting point.
>
> Regarding weaknesses 1 and 2, we acknowledge that the current datasets (Retro* and its superset, USPTO) are indeed noisy and of relatively low quality. This issue serves as a core motivation for our work: to mitigate the impact of poor-quality datasets by integrating expert knowledge. In this study, we leverage expert models, though we aim to extend this approach to incorporate human expertise in the future. This point is also reflected in our updated Table 1. While fine-tuned LLMs adapt to generate outputs closer to the "ground truth," they often fail to produce chemically valid results. In contrast, our proposed methods significantly alleviate the adverse effects of poor-quality data, demonstrating improved chemical validity.
>
> That said, we recognize that our current evaluation still has limitations, and further work is required to expand and refine our assessment methodology.
>
> We perform additional experiments on routes extracted from Pistachio dataset filtered with much more harsh rules to ensure the quality and present it in Appendix A.10.
>
> We appreciate your constructive suggestions and hope our response address your concerns.

---

### Official Review · Reviewer_x71Y · 2024-11-04

**Soundness:** 2
**Presentation:** 2
**Contribution:** 1
**Rating:** 3
**Confidence:** 4

**Summary:**

This paper investigates the potential of pretrained LLM models to engage in in-context learning of synthesis routes when the context includes related reactions from a RAG database.

**Strengths:**

The paper provides a baseline for a straightforward approach to use modern pre-trained AI models in a retrosynthesis context.  The authors explored a variety of different models, demonstrating the practical utility of this approach.

**Weaknesses:**

Given the rapid pace of development, the pretrained models used by the authors are not state-of-the-art at this point in time.  The authors chose not to finetune any model, which limits the exploration of the true potential of the underlying concepts once the models are adapted to handle small molecule chemistry in a reliable fashion.

The price tables listed in this paper for the models are outdated.  GPT-4o mini costs $0.075 per million input tokens and $0.3 per million output tokens, and GPT-4o also costs significantly less than GPT4-turbo cost at the time the authors used it. Providing updated pricing information and evaluation with latest models would improve the evaluation.

The primary metrics of the paper presented in table 1 are not particularly effective in evaluating if the method works in practice.  In particular, the query success rate only indicates if the model learns to copy the target smiles from the prompt and then produce a series of plausibly looking text.  Since the relevance of the reactions is not evaluated, it may be possible that the round-trip validity can be iterated to become accurate by reference to valid SMILES and purchaseable molecules in the database. It remains unclear if the authors ultimately answer the question they posed in their title.

**Questions:**

Could the authors elaborate on why fine-tuning is considered to require substantial resources?  Finetuning of 4o-mini cost $0.15 per million input tokens and was free up to a daily token limit for a couple of months until end of October.  Moreover, finetuning using LORA for publicly available LLMs can be performed with manageable compute costs (the FLOPS required during training without LORA are 6 * params * datatokens, and modern H100 GPU can theoretically reach 1e15 FLOPS/s).  Have the authors considered finetuning on their training set, possibly including additional small molecule SMILES (as others have done), to create a chemically aware LLM and then repeat their current setup?

---

> ### Author Response · Authors · 2024-11-25
>
> Thank you for your insightful feedback on our paper. We have marked in blue the sections addressing your comments, including the addition of evaluation metrics (ROUGE, BLEU, exact match) to measure the similarity between generated routes and ground truth and experiments of finetuning LLM. We believe that query success rate remains a crucial metric, as it reflects the practical availability of our approach. This is particularly important for hard-to-synthesize molecules, where human experts may leverage our system for brainstorming and creative problem-solving.
>
> Regarding your question on fine-tuning: it requires significant resources, including computational power and high-quality data. Retrosynthesis planning is inherently complex, requiring the LLM to handle both expansion and selection, akin to established retrosynthesis frameworks. Current LLMs still face challenges in single-step retrosynthesis prediction (e.g., only a 12% exact match for ChemDFM-13B). While traditional methods rely solely on single-step retrosynthesis models for expansion, our work demonstrates that LLMs can navigate the retrosynthesis space similarly to algorithms like Retro*—though they do not yet surpass its performance.
>
> We fine-tuned ChemDFM-V1.5-8B, which is pretrained on chemistry data and chemically aware. The results, presented in **Appendix A.8**, show that while fine-tuned LLMs achieve comparable ROUGE and BLEU scores, they fail to generate as many chemically valid routes as our approach. This suggests that fine-tuned LLMs may produce text that appears plausible without truly understanding the nuances of retrosynthesis planning.

---

> > ### Comment · Reviewer_x71Y · 2024-12-03
> > **Thank you for the additional comment**
> >
> > I appreciate the answer by the authors and remain skeptical of the practical advance documented in this paper.

---

### Author Response · Authors · 2024-11-25
**Updating of manuscripts**

We sincerely thank all reviewers for their valuable feedback on our manuscript. Their thoughtful comments and suggestions have guided significant improvements to our work. In the revised manuscript, we use a colour-coding system (**blue, orange, and red**) to highlight changes addressing specific comments from Reviewers **x71Y**,**daJj**,**BRau**, respectively.

The key changes included:

- A concrete introduction of experts and knowledge bases in **Section 3**
- Additional evaluation metrics including Rouge, Bleu, exact match in **Section 4.1** and all evaluations afterwards
- Additional baseline methods including EG-MCTS and finetuned ChemDFM in **Section 4.1**
- Clarification of reaction round-trip validity and route validity in **Section 4.1**
- Replacement of misleading “Route RT Validity” in result tables with “Route Validity”
- Expansion of related works on retrosynthesis in **Section 5.1**
- Additional Reproducibility statement after the main text (P.11)
- Additional experiments using a chiral-aware fingerprints (MAP4C) instead of Morgan fingerprints in **Appendix A.8**
- Ablation experiments on finetuned ChemDFM in **Appendix A.9**
- Additional experiments on routes extracted from the Pistachio dataset in **Appendix A.10**

We sincerely thank the reviewers for their constructive feedback that has helped enhance our work.

---

### Note · Authors · 2024-12-03

**Comment:**

We would like to sincerely thank the reviewers for their thoughtful feedback and constructive discussion regarding our submission. The insights provided have been invaluable in helping us refine our work and deepen our understanding of the topic.

After careful consideration, we have decided to withdraw this manuscript. We plan to incorporate the valuable suggestions provided and further develop the work before resubmitting it in the future.

Thank you again for your time and effort in reviewing this submission.

**Withdrawal Confirmation:**

I have read and agree with the venue's withdrawal policy on behalf of myself and my co-authors.